# A randomized, controlled study to assess if allopathic-osteopathic collaboration influences stereotypes, interprofessional readiness, and doctor-patient communication

Ian A. Jones[1]*, Michael LoBasso[2], Johanna Shapiro[3], Alpesh Amin[4]

1 Department Anesthesiology & Pain Medicine, University of Washington, Seattle, Washington, United States of America, 2 Department of Anesthesiology & Perioperative Medicine, UCLA, Los Angeles, California, United States of America, 3 Department of Family Medicine, UCI School of Medicine, Irvine, California, United States of America, 4 Department of Medicine, UCI School of Medicine, Irvine, California, United States of America

* itisianj@gmail.com

**Data Availability Statement:** All relevant data are within the manuscript and its Supporting Information files.

## Abstract

Despite the growing similarities between allopathic (MD) and osteopathic (DO) medical education, few studies have examined allopathic-osteopathic collaboration. The following study focused on stereotypes and student readiness for interprofessional learning. Patient perceptions were also evaluated. Osteopathic and allopathic students were randomly allocated 1:1:1 to work in pairs (MD/DO, MD/MD, DO/DO) at the start of each shift. A questionnaire evaluating student communication was collected from patients at the end of each encounter. Surveys assessing stereotypes and interprofessional readiness were obtained from students at the end of each workday. Data collection was stopped early due to Coronavirus-related safety measures. In the ITT analysis, there were a total of 126 participants (57 students 69 patients). A per-protocol analysis was performed to account for repeat clinic volunteers. No significant differences were detected between student pairs; however, the sensitivity analysis of the questionnaire assessing interprofessional readiness was 8 points higher in the DO/DO group compared to the MD/MD and MD/DO groups (P = 0.0503). In the content analysis of qualitative responses, the MD/DO group was more likely to respond with themes of enjoyment and less concern about stereotypes than the DO/DO group. The MD/DO group was also less likely to report concerns about differences in expectations, methods, and thinking than the MD/MD group. Early trends from this study suggest that DO students may be better positioned to engage in interprofessional learning than their MD counterparts. Additionally, the findings from our content analysis provide evidence that the collaborative experience improved feelings associated with professional legitimacy and credibility among DO students. Taken in aggregate, this study provides justification for a follow-up investigation, as well as a framework for how such studies could best be executed in the future.

**Funding:** Funding for biostatistician consulting services was provided by the UCI Hospital Medicine Program and UC Department of Medicine. The UC Irvine Center for Statistical Consulting is partially supported by grant UL1 TR001414 from the National Center for Advancing Translational Sciences, National Institutes of Health (NIH), through the Biostatistics, Epidemiology and Research Design Unit. The content is solely the responsibility of the authors and does not necessarily represent the official views of the NIH. Additionally, the University of California Medical Humanities Consortium Summer Research Stipend recipient was awarded to Ian Jones and Michael LoBasso for their work on this project. The funder provided support in the form of a summer research stipend for authors I.J. and M.L. but did not have any additional role in the study design, data collection and analysis, decision to publish, or preparation of the manuscript.

**Competing interests:** Alpesh Amin has served as PI or co-I of clinical trials sponsored by NIH/NIAID, NeuroRx Pharma, Pulmotect, Blade Therpeutics, Novartis, Takeda, Humanigen, Eli Lilly, PTC Therapeutics, OctaPharma, Fulcrum Therapeutics, Alexion. He has served as speaker and/or consultant for BMS, Pfizer, BI, Portola, Sunovion, Mylan, Alexion, AstraZeneca, Novartis, Nabriva, Paratek, Bayer, Tetraphase, Achogen LaJolla, PeraHealth, HeartRite, Aseptiscope, Sprightly, Ferring, Spero, Seres, Eli Lilly. All authors declare no conflicts or competing interests for this work. The specific roles of authors are articulated in the 'author contributions' section. These affiliations do not alter authors adherence to PLOS ONE policies on sharing data and materials, and no commercial affiliation played a role in any study-related activities.

# Introduction

Despite a lack of concrete evidence, it is thought that interprofessional collaboration improves patient outcomes by enhancing communication and increasing accessibility to services [1–3]. Numerous studies evaluating collaborative practice/education have been published, but studies to date have largely overlooked collaboration between allopathic (MD) and osteopathic (DO) physicians [4]. This is problematic because osteopathic physicians trained in the United States are licensed medical doctors that have full practice rights. These rights are often retained, even when an osteopathic physician decides to practice in a different county. In contrast, foreign-trained osteopaths are often not licensed to prescribe medications or perform surgeries. As interactions between MD and DO physicians become increasingly commonplace [5], there is increased need to understand how these interactions influence clinical practice and healthcare provider perceptions.

It is unclear whether the quality of patient-physician communication differs among osteopathic and allopathic physicians. Anecdotal accounts and some studies have suggested that osteopathic physicians may have an approach style that is more personal and holistic [6, 7]. However, other studies have provided little evidence that a distinctive approach to physician-patient interactions among osteopathic physicians exists, particularly with respect to time spent with patients [5, 8]. Nevertheless, there are distinctive characteristics of osteopathic medicine that may influence the different communication patterns between osteopathic and allopathic providers [6]. Indeed, in a survey of 3000 osteopathic physicians, 59% believed they practiced differently from allopathic physicians, and 72% of the follow-up responses indicated that factors such as a caring doctor–patient relationship and hands-on style were major distinguishing features [9].

Among the challenges posed by interprofessional collaboration are the stereotypes held by student healthcare practitioners, which may influence their future effectiveness in a multidisciplinary workplace [10]. These stereotypes are of particular concern for osteopathic physicians, as they have historically struggled to gain professional legitimacy and credibility [11]. The following study addressed two principle questions: (1) whether interpersonal collaboration between MD and DO students affects interprofessional stereotypes and readiness; and (2) whether interprofessional collaboration between MD and DO students impacts patient-perceived communication.

# Materials and methods

## Ethics and dissemination of information

All study activities were conducted in accordance with Institutional Review Board (IRB) guidelines for exempt studies. In accordance with IRB guidelines, a formal IRB certification of exemption was requested and provided on 19 June 2019. An information sheet was utilized in lieu of formal Informed Consent. The information sheet and all surveys were administered using REDCap, which allowed participants to enter their deidentified responses directly into the database via iPad. On clinic days where staffing was sufficient to support the predetermined randomization schedule, adult patients aged 18 or older and medical students from two different medical schools (an allopathic school and an osteopathic school) were eligible to participate. Prior to participation, participants (both students and patients) were able to view the Study Information Sheet, which outlined the study's purpose and eligibility criteria. The information sheet was available in both Spanish and English, in accordance with clinic demographics. Participants were prompted to verify that they met the eligibility criteria and indicate their willingness to participate. Information sheet responses were recorded via iPad, and only

students/patients that expressed willingness to participate were enrolled and provided with assessment surveys.

## Data collection

All study activities were conducted at An Lanh Free Medical Clinic in Garden Grove, California. Medical student volunteers are tasked with responsibilities analogous to what would be expected from a medical student during standard clinical training. These responsibilities include, chart checking patients prior to the encounter, independently performing a history and physical exam, and then presenting their assessment and plan to an attending physician. These activities are part of the clinic's standard operating procedure and are not unique to the study itself.

Student surveys were collected at the end of each clinic day. Demographic information included race, ethnicity, sex, whether this was their first time working at the clinic, school year, whether the student was enrolled in a dual degree program, such as a PhD or Master's program/track. Stereotypes were assessed using the Student Stereotypes Rating Questionnaire (SSRQ) [12], a 9-item questionnaire designed to elicit stereotypes between various healthcare professions. Respondents were asked to rate their MD or DO counterparts in terms of their academic ability, interpersonal skills, professional competencies, leadership, how well they worked in a team, independence as a worker, confidence, decision making ability, and practical skills. Interprofessional readiness was assessed using the Readiness for interprofessional learning (RIPLS) Questionnaire, a 10-item questionnaire that measures attitudes toward interprofessional education [13].

Patient questionnaires were obtained while students were presenting the patient to the attending physician. To assess the quality of communication between students and their patients, the 4-item 'Doctors Who Communicate Well' subsection of the Consumer Assessment of Health Plans Study (CAHPS) survey was utilized [14]. Demographic information included race, ethnicity, sex, and whether this was their first time visiting the clinic.

## Statistical analysis

A formal statistical analysis plan was drawn up by the Biostatistics, Epidemiology and Research Design (BERD) unit of UC Irvine's Institute for Clinical and Translational Science (ICTS) Institute prior to the initiation of study activities (S1 Appendix). In brief, the primary analysis was performed on Intention-to-treat (ITT) observations. Multivariate Linear mixed effect model with both student and pair-level random effects was used to assess the difference in SSRQ and RIPLS scores among the three groups. The student-level random effect was intended to account for within student correlation for those students who volunteered at the clinic more than once. The pair-level random effect was used to adjust for the correlation of the responses from the same pair. The model also included the variable of response from MD or DO as a covariate to control for its potential confounding effect. A sensitivity analysis was performed on per-protocol observations using a similar multivariate linear mixed effect model. Additionally, Spearman's correlation test was conducted to assess the association between SSRQ and RIPLS to account for the non-linear association.

In the subset analysis, non-parametric Wilcoxon rank-sum test was used to assess the difference in SSRQ and RIPLS between MD and DO students within the MD/DO pairs. They were compared in both primary and sensitivity analysis. Additionally, a similar Multivariate Linear mixed effect model with pair-level random effect was used to assess the difference in CAHPS scores among the MD/MD, MD/DO, and DO/DO groups. Multivariable linear mixed effect model was also used to assess the association between SSRQ and RIPLS scores from the

students and CAHPS patient experience score. The outcome was CAHPS, while the key explanatory variable was average SSRQ or RIPLS score from the two students in the pair. Similar to the analysis above, within-pair correlation was controlled by adding a pair-level intercept. In the sensitivity analysis, we excluded those pair average responses if either of two responses was not the first-time response.

### Power, and randomization

Power calculation was based on the primary aims of assessing stereotypes and interprofessional readiness, as well as measuring patient experience. Bonferroni correction was used to adjust for multiple comparisons. Since there were 3 primary outcomes (2 student surveys, 1 patient survey), the significance level was adjusted to be 0.05/3 = 0.0167. A one-way ANOVA test was used to calculate the power. Assuming an effect size of 0.33 (variation among the 3 combinations), the design of n = 40 pairs in each combination, totaling n = 120 pairs of medical students, achieved 80% power to detect the differences among the means versus the alternative of equal means using an F test with a 0.0167 significance level. Moreover, the repeated measure design described above was expected to achieve even higher power by collecting multiple data points per pair. The allocation for each combination (MD/DO, MD/MD, DO/DO) was 1:1:1, with randomization to an MD/DO group being treated as the intervention and MD/MD and DO/DO groups being treated as the control. A randomization schedule was provided by the principal biostatistician prior to the initiation of study activities.

### Qualitative data analysis

The study generated three sets of narrative data, which were incorporated into the REDCap surveys: #1) Three words to describe the experience of working with a partner, #2) Challenges of MDs and Dos working together, #3) Strengths and weaknesses of each. A medical educator experienced with qualitative research (author JS) trained a medical student in the theory of qualitative analysis, the development of codes and coding practices, and the interpretation of data [15, 16]. These researchers then analyzed responses from the DO/DO, MD/MD, and DO/MD groups.

The narrative data was analyzed using qualitative methodology, the main goal of which is to discover data meaning. This study relied on both conventional and summative content analysis. We approached data analysis by first reading through all information to get an overall understanding of student responses. Next, we examined specific words and phrases to formulate codes, and generated initial codes, which we then organized into larger thematic categories. The last step used a summative approach to compare and contrast the three subject teams (DO/DO, MD/MD and DO/MD). Each investigator first performed an independent analysis, then worked with their partner to discuss coding differences and reach a consensus summary.

## Results

### Overview

Data collection began Aug 24th, 2019 and terminated March 7th, 2020 due to the cancelation of in-person patient encounters secondary to Coronavirus-related safety measures taken by the clinic. In the ITT analysis, there were a total of 57 student participants and 72 received questionnaires (Fig 1). Students were paired into three groups: 8 MD/DO, 12 MD/MD and 16 DO/DO. As mentioned previously, per-protocol analysis was performed to account for students who had been paired more than once. In the per-protocol analysis, we only included the questionnaire from their first pair assignment for those who were paired multiple times. Additionally, anonymous survey responses were solicited from a total of 91 patients, of whom 69

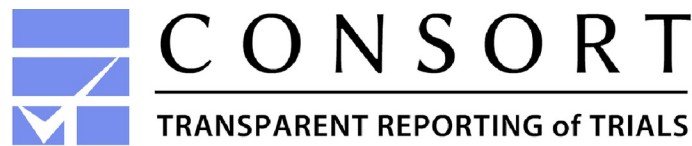

## CONSORT 2010 Flow Diagram

**Enrollment**

Assessed for eligibility (n=75)

Excluded (n=3)
- Not meeting inclusion criteria (n= 0)
- Declined to participate (n=0)
- Other reasons (n=3)

Randomized (n=72)

**Allocation**

Allocated to intervention (n=21)
- Received allocated intervention (n=21)
- Did not receive allocated intervention (n=0)

Allocated to intervention (n=51)
- Received allocated intervention (n=47)
- Did not receive allocated intervention (n=4; did not complete survey correctly, declined to participate)

**Follow-Up**

Lost to follow-up (n=0)

Discontinued intervention (n=0)

Lost to follow-up (n=0)

Discontinued intervention (n=0)

**Analysis**

Analysed (n=21)
- Excluded from analysis (n=0)

Analysed (n=47)
- Excluded from analysis (n=4; did not complete survey correctly)

**Fig 1. Enrolment, intervention allocation, follow-up, and data analysis.**

**Table 1. ITT and sensitivity analysis of SSRQ and RIPLS scores between the MD/MD, MD/DO, and DO/DO groups.**

| SSRQ | | | | | | |
|---|---|---|---|---|---|---|
| | | ITT Analysis | | | Sensitivity Analysis | |
| Group | N | Mean (95% Confidence Interval) | P Value | N | Mean (95% Confidence Interval) | P Value |
| MD/DO | 16 | 42.0 (39.5,44.5) | 0.3039 | 13 | 41.7(38.9,44.6) | 0.3094 |
| MD/MD | 24 | 38.9 (36,41.9) | | 20 | 38.7(35.5,41.8) | |
| DO/DO | 32 | 41.2 (38.4,44) | | 24 | 41.8(38.8,44.8) | |
| RIPLS | | | | | | |
| | | ITT Analysis | | | Sensitivity Analysis | |
| Group | N | Mean (95% Confidence Interval) | P Value | N | Mean (95% Confidence Interval) | P Value |
| MD/DO | 16 | 96(92.1,99.9) | 0.5095 | 13 | 94.1(89.8,98.3) | 0.1372 |
| MD/MD | 24 | 94.5(89.4,99.5) | | 20 | 93.4(88,98.7) | |
| DO/DO | 32 | 99.4(94.6,104.2) | | 24 | 101.7(96.6,106.9) | |

agreed to participate. Among patients surveyed, 18 were treated by MD/DO pairs, 29 were treated by MD/MD pairs and 31 were treated by DO/DO pairs.

Fisher's exact test indicated that there were no demographic differences among the three groups, confirming group randomization was adequately performed. Student and patient demographics are presented as a frequency, with percentage for each group (S2 Appendix).

## Student interprofessional stereotypes and readiness

After controlling whether the response was from MD/DO, no significant differences were detected between student pairs with respect to SSRQ or RIPLS (Table 1). However, in the sensitivity analysis RIPLS was 8 points higher on average in the DO/DO group compared to the other two groups (Table 2). While not significant, the P value for this observation was around the boundary (P = 0.0503). Additionally, there was correlation between SSRQ and RIPLS of approximately 0.44 (S2 Appendix), which is considered a medium-to-large correlation [17].

There was no significant difference in SSRQ and RIPLS between MD and DO responses within the MD/DO pairs. However, in the sensitivity analysis, there was some evidence showing DO students possibly tended to score higher in RIPLS in comparison to MD students with P = 0.07 (Table 3).

## Patient-perceived quality of student communication

There was no significant difference in CAHPS among the three groups, though DO/DO groups did perform better than both MD/DO and MD/MD groups (Table 4). There was also no significant association between CAHPS and SSRQ or RIPLS in both primary and sensitivity analysis (S2 Appendix).

## Qualitative data analysis

Content analysis was used to assess individual responses between MD/MD, MD/DO and DO/DO pairs. When asked to summarize their experience in a few words (Question #1), each

**Table 2. Comparisons of RIPLS in the sensitivity analysis.**

| Comparisons of RIPLS in the Sensitivity Analysis | | |
|---|---|---|
| | Difference Mean (95% Confidence Interval) | P Value |
| MD/DO vs. DO/DO | -7.66 (-14.90, -0.43) | 0.0503 |
| MD/MD vs. DO/DO | -8.37 (-17.82, 1.08) | 0.0973 |

**Table 3. Subset analysis using non-parametric Wilcoxon rank-sum test assessing the difference in SSRQ and RIPLS between MD and DO patients within the MD/DO pairs.**

| | | | | | | |
|---|---|---|---|---|---|---|
| **Subset Analysis on Differences within MD/DO Pairs** | | | | | | |
| **Variable** | **Analysis** | **MD** | | **DO** | | **P Value** |
| | | **N** | **Mean (SD)** | **N** | **Mean (SD)** | |
| SSRQ | ITT | 8 | 40.63 (4.21) | 8 | 43.38 (2.00) | 0.2248 |
| | Per Protocol | 5 | 39.80 (3.90) | 8 | 43.38 (2.00) | 0.0952 |
| RIPLS | ITT | 8 | 94.13 (8.34) | 8 | 97.88 (7.18) | 0.2466 |
| | Per Protocol | 5 | 90.20 (6.10) | | 97.88 (7.18) | 0.0665 |

grouping reported that the experience was collaborative and educational with a similar frequency. However, MD/DO and MD/MD pairs were twice as likely as the DO/DO pairs to respond with themes of enjoyment. When assessing challenges of working together (Question #2), training was identified as the primary barrier, but compared to the DO/DO group, the MD/MD group was more than twice as likely to report concerns about differences in expectations, methods, and thinking potentially leading to difficulty in working together. An equal frequency, but lesser number of responses from each group stated there are no potential barriers. Additionally DO/DO teams were two times more likely than MD/MD to teams to describe stereotypes and perception as a challenge in practice. The MD/DO group fell somewhere between the other two groups in terms of stereotyping concern. Each group had similar responses with respect to perceived strengths and weaknesses (Question #3), regardless of their pairing. All 3 groups reported MD students are more focused on academics and evidence-based medicine, while DO students excel at the physical exam and holistic treatment approaches.

## Discussion and conclusions

This study used a randomized, controlled design to determine how early collaboration between MD and DO students influences interprofessional stereotypes and readiness, as well as whether this collaboration has an effect on patient experience. While the findings are notably weakened by the cessation of the study prior to the study achieving 50% enrollment, several interesting trends did emerge. Among the most notable of these trends was RIPLS, with DO/DO pairs scoring higher with respect to interprofessional readiness and MD/MD pairs scoring lower. Qualitative findings regarding perceived barriers to collaboration in the MD/MD group support this potential lack of readiness. At the same time, the finding that students in the DO/DO group were most likely to mention concerns about negative stereotyping suggests a possible lack of confidence that they will be accepted by MD colleagues. This makes the analysis comparing the MD and DO scores of students when they were paired MD/DO particularly noteworthy. In these pairings, DO students scored better than MD students on the RIPLS questionnaire. Qualitative data also indicated a decrease in concern regarding stereotyping. Taken in aggregate, these findings suggest that, at least early in their career, while DO students

**Table 4. Differences in CAHPS scores among the MD/MD, MD/DO, and DO/DO groups.**

| | | | |
|---|---|---|---|
| **CAHPS** | | | |
| **Group** | **N** | **Mean (95% Confidence Interval)** | **P Value** |
| MD/DO | 18 | 18.9 (18.0, 19.7) | 0.7056 |
| MD/MD | 20 | 18.9 (18.1, 19.7) | |
| DO/DO | 31 | 19.3 (18.6, 19.9) | |

may have concerns about MD stereotyping of their training, they are still better positioned to engage in interprofessional learning than their MD counterparts.

For the SSRQ, which evaluates student stereotypes, there were no clear trends favoring a specific group or student type. However, we did observe a strong correlation between SSRQ and RIPLS, suggesting the two surveys may address similar domains. This finding makes intuitive sense, as individuals harboring negative stereotypes could reasonably be expected to be less ready to engage in interprofessional learning. The apparent increased sensitivity of the RIPLS compared to SSRQ observed in our study could be due to RIPLS capturing the underlying effects of negative interprofessional stereotypes, while also measuring other factors that impact student readiness to engage in interprofessional learning. Whatever the case, the findings suggest that RIPLS may be preferred to SSRQ when evaluating MD/DO perceptions.

One of the major advantages of this study was that quantitative surveys were paired with qualitative questions that allowed for free text responses. DO student responses tended to include themes of enjoyment and were less likely to report stereotypes/perceptions as being a challenge when they were paired MD/DO than when they were paired DO/DO. This suggests that interprofessional collaboration can be used to improve potential negative feelings pertaining to professional legitimacy and credibility among DO students. This finding is in line with anecdotal reports, in which DO physicians have attested to no longer feeling like there is a strong perceived difference between them and their MD counterparts once they enter the workforce where they work collaboratively with MD physicians [18]. Indeed, while differences exist between the osteopathic and allopathic curricula [19], the same basic science knowledge is expected for DO and MD students [20]. Given that rank and file osteopathic practitioners have struggled (at least historically) with professional identification among their MD counterparts [9, 21], providing early exposure to the collaborative environment could be an important tool for addressing potential stigma [22].

This study has several limitations. As mentioned previously, the study was terminated prior to completion of predetermined enrollment due to the cancelation of in-person patient encounters secondary to Coronavirus-related safety measures taken by the clinic. As a result of early study termination, there was insufficient data to draw strong statistical conclusions from the survey data, particularly with respect to how interprofessional learning may influence patient perception. Nevertheless, it should be noted that DO/DO groups performed the best on patient communication, which is consistent with prior studies which have found that the communication style of osteopathic physicians may better incorporate aspects such as family, social activities, and patient emotions [6]. It is also consistent with studies that have suggested that an interest in osteopathic techniques may be associated with higher empathy scores [23]. There may also be a limitation or inherent bias that occurred as a result of student self-selection, as the study was conducted at a clinic run by volunteers. As a result, the population of students studied may have personality traits that differ slightly from the student body at large. Another limitation of this study was that several students volunteered at the clinic multiple times; however, this limitation was accounted for using pair-level random effects. One major confounder inherent to this study that could not be controlled for was that students from only two institutions participated in the study. While this limitation was unavoidable, as it is relatively unique for a clinic to be run by both MD and DO students, it does highlight the need for additional opportunities that would allow MD and DO students to have collaborative experiences early in their medical career, particularly given the potential benefits observed in our limited sample.

Taken in aggregate, the framework and preliminary data obtained in this study support further investigation. The quantitative findings may be of particular value when selecting outcome instruments and determining enrollment for follow-up studies. Additionally, the

preliminary qualitative analysis suggests that future collaborative experiences could be established and studied safely, while having a neutral-to-positive effect on participants. Potential directions for future studies include following students longitudinally (e.g., by measuring scores pre- and post-intervention) or comparing the relative effectiveness of formal didactic education (which could include a briefing and/or debriefing component) to that of early clinical exposure alone. It is also possible that a combined approach that included both a formal didactic component and real-world exposure might have a synergistic effect.

## Supporting information

**S1 Data.**
(XLSX)

**S1 Appendix. Statistical analysis plan.**
(DOCX)

**S2 Appendix. Additional study data.** Table data broken up into excel tabs corresponding to demographic data (B1); the correlation between SSRQ and RIPLS (B2); association between CAHPS and the SSRQ and RIPLS questionnaires (B3).
(XLSX)

## Acknowledgments

The authors would like to thank Yanjun Chen, the principal biostatistician responsible for helping develop the statistical analysis plan, power, and randomization. We would also like to thank An Lanh Free Medical Clinic, their staff, and volunteers for making this project possible. Additionally, the authors would like to give a special thanks to the following: Calvin Yichih Xu <calvinyx@uci.edu> for logistical support, Baotran Vo, MD bnvo1@hs.uci.edu for her enduring commitment to community health and education at An Lanh, and Dai Nguyen <dain1@hs.uci.edu> for her insight and feedback early in the study drafting process.

### Presentations

Preliminary results were presented at the 2020 USC Innovations in Medical Education Conference.

## Author Contributions

**Conceptualization:** Ian A. Jones, Michael LoBasso, Johanna Shapiro, Alpesh Amin.

**Data curation:** Ian A. Jones, Michael LoBasso, Johanna Shapiro.

**Formal analysis:** Ian A. Jones, Michael LoBasso, Johanna Shapiro.

**Funding acquisition:** Ian A. Jones, Michael LoBasso, Alpesh Amin.

**Investigation:** Ian A. Jones, Michael LoBasso, Johanna Shapiro, Alpesh Amin.

**Methodology:** Ian A. Jones, Michael LoBasso, Johanna Shapiro, Alpesh Amin.

**Project administration:** Ian A. Jones, Michael LoBasso.

**Resources:** Ian A. Jones, Michael LoBasso, Alpesh Amin.

**Software:** Ian A. Jones, Michael LoBasso.

**Supervision:** Ian A. Jones, Michael LoBasso, Johanna Shapiro, Alpesh Amin.

**Validation:** Ian A. Jones, Michael LoBasso.

**Writing – original draft:** Ian A. Jones, Michael LoBasso, Johanna Shapiro, Alpesh Amin.

**Writing – review & editing:** Ian A. Jones, Michael LoBasso, Johanna Shapiro, Alpesh Amin.

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
