## [Decision Letter · Decision Letter 0]

15 Jun 2022

PONE-D-21-32629A Randomized, Controlled Study to Assess how Allopathic-Osteopathic Collaboration Effects Stereotypes, Interprofessional Readiness, and Doctor-Patient CommunicationPLOS ONE

Dear Dr. Jones,

Thank you for submitting your manuscript to PLOS ONE. After careful consideration, we feel that it has merit but does not fully meet PLOS ONE’s publication criteria as it currently stands. Therefore, we invite you to submit a revised version of the manuscript that addresses the points raised during the review process.

 Please note that we have only been able to secure a single reviewer to assess your manuscript. We are issuing a decision on your manuscript at this point to prevent further delays in the evaluation of your manuscript. Please be aware that the editor who handles your revised manuscript might find it necessary to invite additional reviewers to assess this work once the revised manuscript is submitted. However, we will aim to proceed on the basis of this single review if possible.  Your manuscript has been assessed by an expert reviewer, whose comments are appended below. The reviewer has highlighted concerns about several aspects of the reporting and study design. Please ensure you respond to each point carefully in your response to reviewers document, and modify your manuscript accordingly.

We look forward to receiving your revised manuscript.

Kind regards,

Joseph Donlan

Editorial Office

PLOS ONE

Journal Requirements:

2. During your revisions, please confirm whether the wording in the title is correct and update it in the manuscript file and online submission information if needed. Specifically, alter the title from "A Randomized, Controlled Study to Assess how Allopathic-Osteopathic Collaboration Effects Stereotypes, Interprofessional Readiness, and Doctor-Patient Communication" to "A Randomized, Controlled Study to Assess how Allopathic-Osteopathic Collaboration Affects Stereotypes, Interprofessional Readiness, and Doctor-Patient Communication""""

3. You indicated that ethical approval was not necessary for your study. We understand that the framework for ethical oversight requirements for studies of this type may differ depending on the setting and we would appreciate some further clarification regarding your research. Could you please provide further details on why your study is exempt from the need for approval and confirmation from your institutional review board or research ethics committee (e.g., in the form of a letter or email correspondence) that ethics review was not necessary for this study? Please include a copy of the correspondence as an ""Other"" file."""

4. Thank you for stating the following in the Competing Interests/Financial Disclosure section:

“The University of California Medical Humanities Consortium Summer Research Stipend recipient was awarded to Ian Jones and Michael LoBasso for their work on this project.

Alpesh Amin reported serving as PI or co-I of clinical trials sponsored by NIH/NIAID, NeuroRx Pharma, Pulmotect, Blade Therpeutics, Novartis, Takeda, Humanigen, Eli Lilly, PTC Therapeutics, OctaPharma, Fulcrum Therapeutics, Alexion. He has served as speaker and/or consultant for BMS, Pfizer, BI, Portola, Sunovion, Mylan, Alexion, AstraZeneca, Novartis, Nabriva, Paratek, Bayer, Tetraphase, Achogen LaJolla, PeraHealth, HeartRite, Aseptiscope, Sprightly.”

We note that one or more of the authors are employed by a commercial company:

5. We note that you have stated that you will provide repository information for your data at acceptance. Should your manuscript be accepted for publication, we will hold it until you provide the relevant accession numbers or DOIs necessary to access your data. If you wish to make changes to your Data Availability statement, please describe these changes in your cover letter and we will update your Data Availability statement to reflect the information you provide

Reviewers' comments:

Reviewer's Responses to Questions

**Comments to the Author**

1. Is the manuscript technically sound, and do the data support the conclusions?

Reviewer #1: Partly

2. Has the statistical analysis been performed appropriately and rigorously? 

Reviewer #1: I Don't Know

3. Have the authors made all data underlying the findings in their manuscript fully available?

Reviewer #1: Yes

4. Is the manuscript presented in an intelligible fashion and written in standard English?

Reviewer #1: Yes

5. Review Comments to the Author

Reviewer #1: The authors present a study about differing perceptions of DO and MD physicians. I have some concerns about the methodology and lack of detail in the paper.

There is a need for some upfront explanation of the DO/MD distinction – the term osteopathic physician is not universally understood (I think it is a US-centric system that does not exist in other countries). In other countries, osteopath would not normally be regarded as a ‘physician’. These titles and the associated roles need to be better described in the introduction.

Some further detail on the methods would be useful, for example:

• The participants are described as students ‘volunteering’ in the clinic – in what circumstances do students volunteer? Are these formal clinical placements? Was the task (patient interview and examination) part of routine training or an additional task specific to this study?

• Who was responsible for recruiting students and patients? How were patient participants identified and recruited?

• Were all the students from the same medical school/s? What is the curriculum structure in terms of patient communication training?

The qualitative data items are described very briefly only – are these questions incorporated into the surveys? This needs further explanation.

To what extent is the pairing for the activity likely to have impacted on perceptions? Is it possible that the individuals within each pair had differing views even before the task? This is more likely I think, given that there seems to have been no briefing or preparation of the pairs prior to the task.

The design of this study seems to be somewhat flawed in that it seems to assume that simply doing a task together is influencing perceptions – I would think that as an educational opportunity, expanding the task to including briefing and debriefing components would be more useful and more likely to generate differences/educational benefit.

In terms of the impact of the study in adding knowledge to the field, my previous comment reflects a concern I have about the overall value of this study.

Some specifics:

• Acronyms used in the abstract need to be given in full in the first instance

• Language in the abstract needs to be refined – first section reads awkwardly

• Page 5 line 67 should read ‘a’ rather than ‘an’

6. PLOS authors have the option to publish the peer review history of their article (what does this mean?). If published, this will include your full peer review and any attached files.

Reviewer #1: No

---

## [Author Response · Author response to Decision Letter 0]

29 Jun 2022

Per the instructions provided, responses have been uploaded as a separate file labeled 'Response to Reviewers"

---

## [Decision Letter · Decision Letter 1]

5 Oct 2022

PONE-D-21-32629R1A Randomized, Controlled Study to Assess if Allopathic-Osteopathic Collaboration Influences Stereotypes, Interprofessional Readiness, and Doctor-Patient CommunicationPLOS ONE

Dear Dr. Jones,

Thank you for submitting your manuscript to PLOS ONE. After careful consideration, we feel that it has merit but does not fully meet PLOS ONE’s publication criteria as it currently stands. Therefore, we invite you to submit a revised version of the manuscript that addresses the points raised during the review process.

Your manuscript has been evaluated by two reviewers and their comments are attached below. They applaud your efforts to strengthen the manuscript, but mention a few minor points that should still be addressed. This includes a better integration of current literature into your discussion.Could you please revise your manuscript to address all their concerns?

We look forward to receiving your revised manuscript.

Kind regards,

Thomas Tischer

Staff Editor

PLOS ONE

Journal Requirements:

Reviewers' comments:

Reviewer's Responses to Questions

**Comments to the Author**

1. If the authors have adequately addressed your comments raised in a previous round of review and you feel that this manuscript is now acceptable for publication, you may indicate that here to bypass the “Comments to the Author” section, enter your conflict of interest statement in the “Confidential to Editor” section, and submit your "Accept" recommendation.

Reviewer #1: All comments have been addressed

Reviewer #2: (No Response)

2. Is the manuscript technically sound, and do the data support the conclusions?

Reviewer #1: Yes

Reviewer #2: Partly

3. Has the statistical analysis been performed appropriately and rigorously? 

Reviewer #1: I Don't Know

Reviewer #2: Yes

4. Have the authors made all data underlying the findings in their manuscript fully available?

Reviewer #1: Yes

Reviewer #2: Yes

5. Is the manuscript presented in an intelligible fashion and written in standard English?

Reviewer #1: Yes

Reviewer #2: Yes

6. Review Comments to the Author

Reviewer #1: The authors have effectively addressed my earlier concerns and the paper reads well. I am satisfied that the paper is of sufficient quality for publication.

Reviewer #2: Thank you for the opportunity to review this revision of the manuscript. Overall, the manuscript reads satisfactorily and the changes have strengthened the manuscript. There needs to be greater links to the literature in the Discussion/Conclusion section. Much of the commentary here is no referenced or linked back to the literature.

Some minor additional changes will help the reader better understand the work:

Page 5, line 85 - it would be good to separate into paragraphs the data collected from the students and that collected from the patients.

Page 6 - line 94- clarify that the Student Stereotypes Rating Questionnaire measures stereotypes of one's own profession and of other health professions. This isn't entirely clear at present.

Page 6 - line 103, provide an example of a dual degree

Page 7, line 131 - who was the control group for the study? This isn't clear from the description provided.

Page 9 - line 152 - please provide additional data about the thematic analysis, particularly who performed the analysis, their experience with the analytical approach etc. The COREQ (https://www.equator-network.org/reporting-guidelines/coreq/) will be helpful in guiding the additional detail required. On reviewing the Results, it would appear that these are more consistent with a content analysis approach given the commentary on frequency etc. (e.g. "compared to the DO/DO group, the MD/MD group was more than twice as likely to report concerns"

Page 13, line 227 - may be better to describe the MD/MD combination as demonstrating lower RIPLS scores rather than 'worst'

7. PLOS authors have the option to publish the peer review history of their article (what does this mean?). If published, this will include your full peer review and any attached files.

Reviewer #1: No

Reviewer #2: **Yes: **Brett Vaughan

---

## [Author Response · Author response to Decision Letter 1]

12 Oct 2022

Please see line-item responses to review concerns below. Overall, we believe the suggested edits improved the quality of the manuscript. We thank both the editorial office and reviewers for their time and consideration.

Thank you for the opportunity to review this revision of the manuscript. Overall, the manuscript reads satisfactorily and the changes have strengthened the manuscript. There needs to be greater links to the literature in the Discussion/Conclusion section. Much of the commentary here is no referenced or linked back to the literature.

In an effort to reduce manuscript bloat, our discussion focused on contextualizing findings, rather than making commentary on the literature as a whole. There is also a paucity of data on MD/DO interaction in the literature, as evidenced by the delays in finding qualified reviewers for this manuscript. Taken together, these factors lead to a discussion with less citations than might otherwise be included in a prospective study of similar quality. It should also be noted that, the original manuscript write up had contained close to twice as many citations. However, after critical appraisal, many of these references were removed, as they contained questionable methods and/or provided low quality data. Nevertheless, we understand the reviewers' concern and have attempted to add additional citations to our discussion. In total 6 new references have been added to the discussion section of the manuscript. 

Page 5, line 85 - it would be good to separate into paragraphs the data collected from the students and that collected from the patients.

The requested change has been made

Page 6 - line 94- clarify that the Student Stereotypes Rating Questionnaire measures stereotypes of one's own profession and of other health professions. This isn't entirely clear at present.

The requested change has been made

Page 6 - line 103, provide an example of a dual degree

The requested change has been made

Page 7, line 131 - who was the control group for the study? This isn't clear from the description provided.

Control was not discussed in this section. This information has been added. In brief, the allocation to work in an MD/DO group was the intervention, and allocation to work in the MD/MD or DO/DO group was the control. Generally speaking, MD and DO students don’t have a chance to work together during the first two years of their medical education. Thus, allocation to an MD/DO group during an early medical education represents a deviation from standard practice.

Page 9 - line 152 - please provide additional data about the thematic analysis, particularly who performed the analysis, their experience with the analytical approach etc. The COREQ (https://www.equator-network.org/reporting-guidelines/coreq/) will be helpful in guiding the additional detail required. On reviewing the Results, it would appear that these are more consistent with a content analysis approach given the commentary on frequency etc. (e.g. "compared to the DO/DO group, the MD/MD group was more than twice as likely to report concerns"

The reviewer may be correct that in the end our analysis ended up being more content than thematic. Although, the authors believe that a case could be made for our use of the term “thematic analysis”, particularly given the fact that, in an inductive approach, we were able to generate themes based on the preliminary coded data. However, in the interest of addressing the reviewers concern, we have re-written the qualitative methods section to put what we did in a conventional and summative content analysis framework. 

Regarding COREQ, which is a 32 item checklist, we believe that it would likely not be possible or constructive to employ given the limited scope of the data. The qualitative component was a small part of this study meant to add context to the quantitative data obtained through randomization and the use of validated questionnaires. 

We have also tried to address concerns regarding the experience of the individuals performing the analysis. However, this concern is a bit hard to address without unblinding and/or including the CV of author JS. The manuscript now states explicitly which author led this portion of the analysis. We are also happy to elaborate further if the reviewer/editor believes that additional information about qualifications is needed.

Page 13, line 227 - may be better to describe the MD/MD combination as demonstrating lower RIPLS scores rather than 'worst'

The requested change has been made

---

## [Editor Report · Decision Letter 2]

11 Nov 2022

A Randomized, Controlled Study to Assess if Allopathic-Osteopathic Collaboration Influences Stereotypes, Interprofessional Readiness, and Doctor-Patient Communication

PONE-D-21-32629R2

Dear Dr. Jones,

We’re pleased to inform you that your manuscript has been judged scientifically suitable for publication and will be formally accepted for publication once it meets all outstanding technical requirements.

Kind regards,

Nabeel Al-Yateem, PhD

Academic Editor

PLOS ONE
---

## [Editor Report · Acceptance letter]

23 Nov 2022

PONE-D-21-32629R2 

A Randomized, Controlled Study to Assess if Allopathic-Osteopathic Collaboration Influences Stereotypes, Interprofessional Readiness, and Doctor-Patient Communication 

Dear Dr. Jones:

I'm pleased to inform you that your manuscript has been deemed suitable for publication in PLOS ONE. Congratulations! Your manuscript is now with our production department. 

Kind regards, 

on behalf of

Dr. Nabeel Al-Yateem 

Academic Editor

PLOS ONE